# AFsample2 predicts multiple conformations and ensembles with AlphaFold2
Yogesh Kalakoti & Björn Wallner ✉

Understanding protein dynamics and conformational states is crucial for insights into biological processes and disease mechanisms, which can aid drug development. Recently, several methods have been devised to broaden the conformational predictions made by AlphaFold2 (AF2). We introduce AFsample2, a method using random MSA column masking to reduce co-evolutionary signals, enhancing structural diversity in AF2-generated models. AFsample2 effectively predicts alternative states for various proteins, producing high-quality end states and diverse conformational ensembles. In the OC23 dataset, alternate state models improved ($\Delta$TM>0.05) in 9 out of 23 cases without affecting preferred state generation. Similar results were seen in 16 membrane protein transporters, with 11 out of 16 targets showing improvement. TM-score improvements to experimental end states were substantial, sometimes exceeding 50%, improving from 0.58 to 0.98. Additionally, AFsample2 increased the diversity of intermediate conformations by 70% compared to standard AF2, producing highly confident models potentially representing intermediate states. For four targets, predicted intermediate states were structurally similar to known structural homologs in the PDB, suggesting that they are true intermediate states. These findings indicate that AFsample2 can used to provide structural insights into proteins with multiple states, as well as potential paths between the states.

Proteins are the workhorses of life, serving as the building blocks of cells and playing crucial roles in almost every biological process. They exhibit a wide range of functions, including catalyzing biochemical reactions, providing structural reinforcement, and even acting as conduits in intracellular communication[1]. Proteins adopt intricate three-dimensional configurations, often existing within structural ensembles that exhibit various states, collective movements, and dynamic fluctuations, all essential for executing their function[2,3]. Processes such as folding, signal transduction, enzyme catalysis, and molecular recognition are all driven by the type and extent of structural dynamics associated with the protein system. Conventional experimental structural biology methods such as X-ray crystallography and cryogenic electron microscopy can provide a few highly accurate snapshots of the overall conformational ensemble of the protein system[4–6]. However, these snapshots only represent a fraction of possible states and have to be supplemented by molecular dynamics (MD) or other similar solutions to infer molecular mechanisms. Furthermore, computational costs related to MD at biologically relevant timescales are not viable in practice. Other

experimental methods such as Nuclear Magnetic Resonance could potentially profile the dynamic nature of the protein molecule but are limited by scale[7].

Recent advancements in in-silico protein structure determination have largely been an outcome of intelligent data processing and generative artificial intelligence. Methods like AlphaFold2[8] (AF2) and RosettaFold[9] have demonstrated exceptional levels of success in determining accurate protein structure from evolutionary sequence information provided as multiple sequence alignments (MSAs). However, the default versions of these workflows are trained to estimate a single high-confident model of the structure of a protein. This is a limitation since the entire conformational landscape has to be considered in order to get insights into the mechanistic basis of protein function. Therefore, an ideal sequence-to-structure prediction system should have the ability to model the entire conformational ensemble for a given protein, identify states, and trace physically viable paths in estimated ensembles. Our recently developed AFsample method[10] captured different conformations of multimeric proteins by increasing the sampling rate and introducing noise by enabling dropout layers at inference. The method achieved state-of-the-art performance and was one of the top-

Division of Bioinformatics, Department of Physics, Chemistry and Biology, Linköping University, Linköping, Sweden. ✉e-mail: bjorn.wallner@liu.se

ranked at CASP15[11] (2022). Additional strategies have also been proposed to induce conformational diversity in AF2 predictions by subsampling the MSA, using shallow MSAs[12], in-silico alanine scan as in SPEACH_AF[13], introducing alanines at random positions in target sequence before MSA search[14], or clustering the MSA as in AFcluster[15]. All of these methods work by effectively reducing the information to AF2 to allow the system to explore alternative solutions. In addition, combining shallow MSAs with the use of templates from PDB[16] or MD has also been explored to investigate conformational landscapes using AF2[17].

In this work, we present AFsample2, which employs MSA column masking by randomly changing columns to "X" to diminish the constraints exerted by co-evolutionary signals in MSA. Thereby increasing the

structural heterogeneity of models generated with the AF2 neural network. As depicted in Fig. 1a, Traditional methods retain the information on co-variation, which in turn constrains the inference system to generate similar structures. AFsample2, on the other hand, removes those constraints by masking columns in the MSA to partially remove some covariance information, thereby increasing the chances of generating alternate conformations. While MSA column masking has been used by other methods, AFsample2 is the first method that has integrated the masking procedure into the AlphaFold code, which enables the generation of models using different MSA masking for every single model with no additional overhead or pre-processing. Thus, each model generated by AFsample2 is generated with a unique randomly masked MSA. AFsample2 was able to improve the

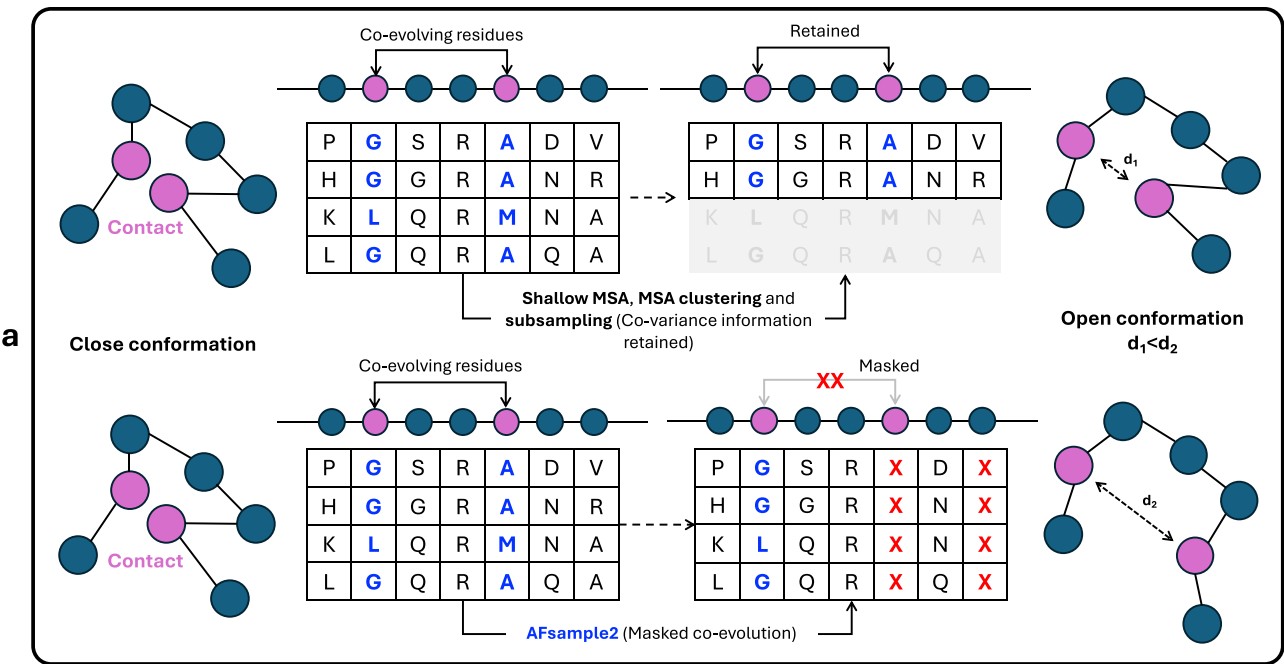

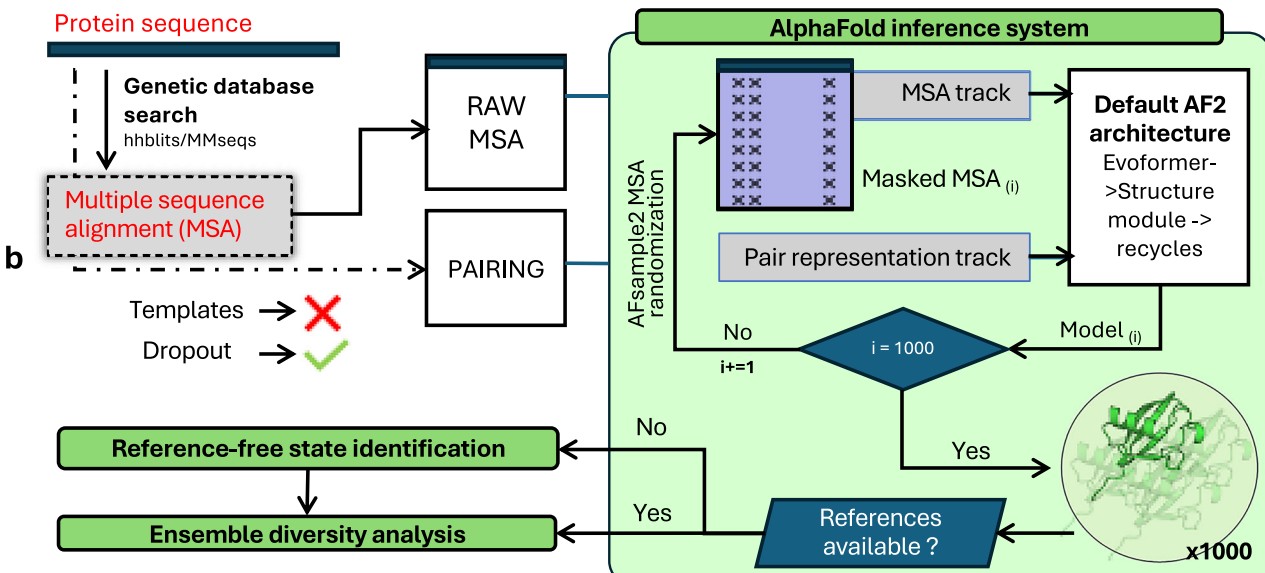

**Fig. 1 | Summary of the AFsample2 workflow. a** Two strategies to alter the MSA, (top) MSA subsampling act on the rows of the MSAs, (bottom) column masking, break co-evolving residues, and potentially contact networks. **b** AFsample2 pipeline starts by generating MSAs for a given protein sequence. This is followed by randomized MSA masking (replacing a % of columns with X, denoting "unknown

residue") in a way such that a unique MSA profile is fed into the system at every instance of the inference run. The generated ensemble is either passed to the diversity analysis workflow or the state-identification workflow, depending on the availability of reference states.

prediction of alternative states for a wide range of proteins. The improvement was quantified based on the ability of the inference system to generate high-quality end states and diverse conformational ensembles. The models for, in particular, the alternate state, are substantially improved for 9/23 cases when compared to the vanilla AF2 system (called AFvanilla thereafter) in the open-closed dataset (OC23) while being at par for other cases, never deteriorating. The performance is maintained on an additional set of 16 membrane protein transporters, with the alternate state substantially improved for 11/16 targets. The improvement as measured by TM-score to experimental end states is sometimes massive with improvements over 50%, basically going from mediocre TM-scores of 0.58 to almost perfect 0.98. However, the improvements are not only in the end states, AFsample2 also improves the diversity by generating 70% more conformations between the end states when compared to the AFvanilla. Some of these intermediate conformations can be mapped to known structures, indicating that they might actually be accurate on-path representations of conformations between states.

In summary, this study presents significant methodological improvements that enhance the capability of MSA-based generative models to capture the conformational landscape of a given protein system.

## Results

### Method development

The primary objective of this study was to improve the sampling of conformational states by introducing more noise than simply turning on the dropout layers at inference[10]. In AFsample2, the noise is introduced by randomly masking columns in the MSA by "X" (denoting unknown residue), with the rationale of breaking covariance constraints in the MSA, see Fig. 1a. By breaking covariance signals, the inference system is allowed to explore and arrive at different solutions for the given protein, ultimately increasing the diversity of the generated protein ensemble. A similar strategy for introducing noise to MSAs has previously been attempted with SPEACH_AF[13], where a sliding window of alanines (instead of "X") was introduced at specific columns in the MSA to break interacting residues. Although sometimes effective, this strategy is dependent on in-silico mutagenesis of the MSA, requiring prior knowledge of the interacting residues to guide this process. Thus, SPEACH_AF relies on having either experimental structures or that models can be predicted for one of the states. AFsample2 does not have this limitation since the column-based masking is completely unbiased and provides a general solution without the need for any additional information.

### Overview of the AFsample2

In summary, given a protein sequence, AFsample2 follows a four-step process to generate diverse protein structures using a modified version of the AF2 inference system. It starts by (i) querying sequence databases to generate MSAs, (ii) Randomly masking MSA columns with "X" using a pre-defined probability (e.g., 15%), but not the first row in the MSA, which is the target sequence (iii) running inference on a uniquely randomly masked MSA for each model with dropout activated and lastly, (iv) depending on the availability of reference states, identifying state representatives with clustering, confidence and extremity selection, followed by ensemble analysis. A schematic representation of the workflow is summarized in Fig. 1b.

### Effect of MSA masking on generated models

The amount of MSA masking, i.e., the fraction of randomized positions, was observed to be the most important factor in the ability of the inference system to generate alternate conformational states. It was observed that increasing the MSA masking increased the chances of generating end-state conformations for a given protein. This trend is summarized in Fig. 2a (left), where MSA masking generates significantly better models compared to no masking (0%) for the alternate state (open in these cases) across a set of diverse proteins with well-defined open and closed conformations (the OC23 set, see "Methods"). The aggregate TM-score for the best alternate (open) conformation increases from 0.80 for no masking to 0.88 with

15% masking while showing a marginal improvement from 0.89 to 0.90 for the closed conformation. Beyond 30% masking, performance drops first for the open conformations and subsequently for the closed conformations.

In addition, it has previously been reported that the model confidence of AF2 predictions deteriorates with increased subsampling[18]. A similar trend was also observed here, where the mean confidence gradually decreased with increasing MSA masking (Fig. 2a (right)). The decrease is linear from 0% to 35% with a 2% drop in confidence for every 5 percentage points of masking, followed by a rapid drop in model confidence beyond 35% masking. Since MSA masking essentially removes information, this trend is expected. However, it is important to realize that the decrease in model confidence up to 20% masking is not coupled to lower quality models. It is most likely an effect that the mask itself renders more uncertainty in the prediction, which in turn results in lower model confidence. Overall, 15% randomization seems to perform marginally better than other settings. However, by analyzing the per-target performance (Fig. 2b), it can be seen that different levels of masking yield the best performance for different target proteins, e.g., 20% masking generates the best model for P40131, while 5% masking is optimal for P71147. The best TM-scores for each protein at various masking levels are visualized in Fig. 2b. Even though the optimal magnitude of masking might differ between targets, it is true that in most cases, masking is always better than no masking for the same level of sampling. For comparative analysis and simplicity, AFsample2 using 15% masking was used for the downstream analysis. However, we do stress that for real applications, the amount of masking should be sampled for optimal performance.

### The importance of sampling

It has been previously established that increased sampling improves the chances of generating alternative conformations[10,19]. But the question is, how much sampling is enough? To answer this, we estimated the best TM-score for the open and closed states, respectively, as the number of samples increased and for different levels of masking 0–50%, see Fig. 2c. Indeed, generating more sampling increases the chances of generating higher-quality models for all levels of masking. The improvement is larger for predicting the open conformation, reflecting the fact that AF2 has a preference for predicting the closed conformation in this case, leaving more room for improvement of the open conformation. The increasing trend is most pronounced for fewer samples, reflecting the switch from no sampling to actual sampling, but it is still increasing even up to 1000 samples, indicating that sampling more is always better. However, considering the trade-off between speed and performance, 1000 samples at 15% masking is a reasonable default.

### No diversity for targets with minor state differences

While it is important to introduce large diversity for targets with large differences between the states, it is equally important that the diversity is not introduced in targets with smaller or no differences between the states. To evaluate this, we compared the distribution of TM-scores to the open and closed state for models generated by AFsample2, for OC23 and OC > 85, a set with targets that have TM-score between open and closed in the [0.85,1] range, see Fig. S1. There is a clear trend that targets with large state differences exhibit wider distributions of TM-scores and that targets with small state differences have a narrow distribution. Thus, AFsample2 does not introduce diversity in general, but only when it is necessary.

### Comparing AFsample2 with existing methods to generate alternative conformations

The performance of AFsample2 was compared to standard AF2 (AFvanilla), MSAsubsample[12], AFcluster[15], SPEACH_AF[13], and AFsample[11], by generating 1000 models for each protein in the OC23 dataset. It should be noted that all experiments have been conducted at the same level of sampling for a fair evaluation. For instance, in the case of SPEACH_AF, instead of using the default 15 models per alanine masked MSA, the factor was increased such that 1000 models in total were generated. Fig. 3a shows the best-generated

**Fig. 2 | Overall summary and analysis of MSA randomization strategy in AFsample2.**
**a** Effectiveness of the MSA randomization strategy in terms of generating high-quality models and aggregate confidence for both open and closed states, error bars: standard error, *n* = 23. **b** Per target best open and closed models for different MSA randomizations. **c** Highest TM-scores for open and close conformation with number of samples generated. Sampling more models increases the chances of generating better models, and is significantly more potent with the proposed randomizations.

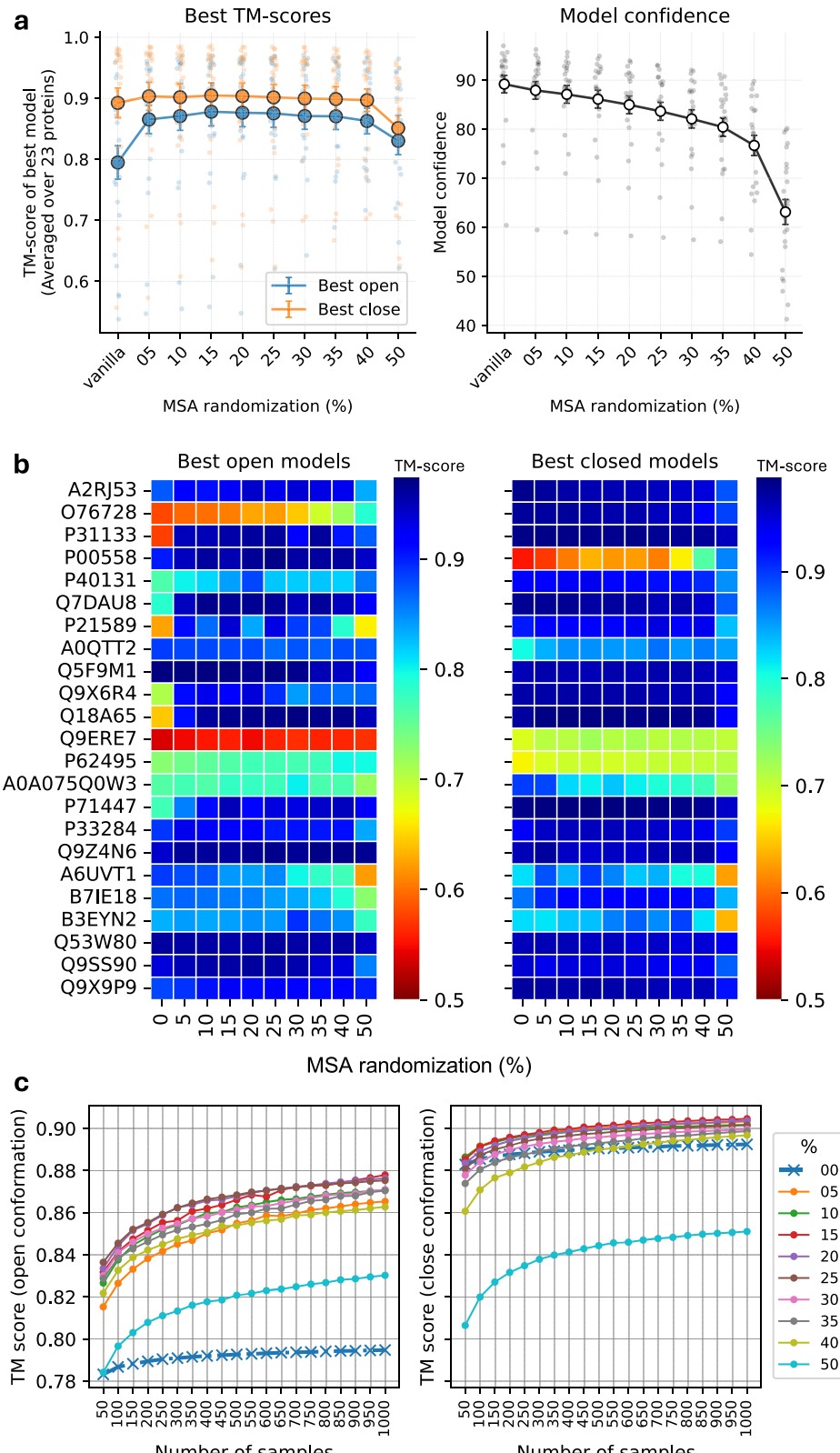

models in the ensemble for the two states as measured by TM-score for each method, and Fig. 3b shows the distribution of the best models. Individual TM-scores for open and closed states for all targets and methods are shown in Fig. S2. While nearly all methods successfully predict the closed state, with all but three targets achieving TM-scores > 0.8 across all methods, the prediction of the open state shows much greater variability. In this case, the open-state models generated by AFsample2 are significantly better than those produced by the other methods, with *P* < 0.01 for MSAsubsample and SPEACH_AF and *P* < 0.001 to the other methods using Wilcoxon-signed-rank test (Fig. 3b (left)).

The performance of all benchmarked methods to consistently produce good models of both states are summarized in the form of an AUC plot (Fig. 3c) where the level of success was quantified as a fraction of total targets for which both states were present in the generated ensemble for different

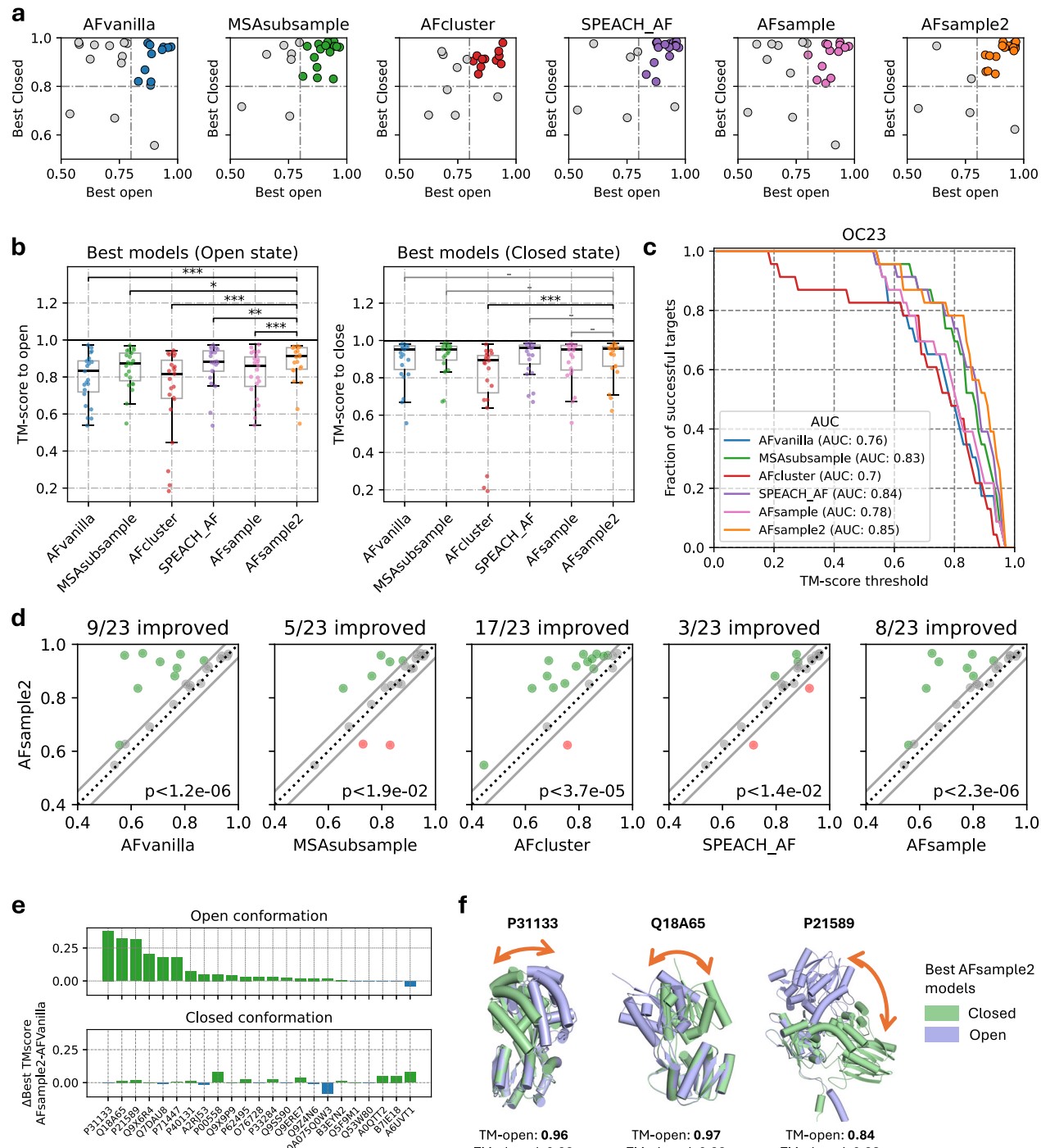

**Fig. 3 | AFsample2 demonstrates a higher ability to generate good-quality open and closed conformations than baseline methods. a** Comparing TM-scores of best models generated for open and closed states by all methods. **b** Best-generated models for the OC23 dataset by all methods under consideration in terms of similarity (TM-score) to the reference states show significant (Wilcoxon-signed-rank test *p* value: *<0.05, **<0.01, ***<0.001, − ns, *n* = 23) improvement, especially for the open state. **c** Fraction of successful targets compared among methods at different levels of TM-score thresholds. Success: TM-score > threshold (for both states). **d** comparing the minimum of the best two selections (one for each state) for all methods against AFsample2 |Δ|TM < 0.05 in gray, green AFsample2 improved, red AFsample2 worsen (Wilcoxon-signed-rank test *p* value, *n* = 23). **e** Improvement of AFsample2 over AFvanilla for open and closed conformations. **f** Examples illustrating generated states by AFsample2 along with their similarity to reference pdbs.

TM-score thresholds for success. It is evident from the AUC plot that AFsample2 consistently performs better than most other methods. AFsample2 generates models of both states with TM-score > 0.8 for 78.3% of the targets, while AFvanilla, MSAsubsample, AFcluster, SPEACH_AF, and AFsample do it for 47.8%, 69.6%, 47.8%, 73.9%, and 56.5% of the targets respectively. We observe a marginal improvement over MSAsubsample and

SPEACH_AF. The latter is unexpected since AFsample2 is essentially a random, unbiased baseline to SPEACH_AF, and the conditions for SPEACH_AF are also ideal for this dataset since the closed state is perfectly predicted by AFvanilla and can thus be used in SPEACH_AF to effectively bias the breaking of contacts. Despite these favorable conditions, SPEACH_AF does not perform better than randomly masking the columns.

Furthermore, we also compared the generation of models for each method against AFsample2 using the minimum TM-score for the two states as a measure of performance (Fig. 3d). In almost all cases, models generated by AFsample2 demonstrate higher ($\Delta$TM > 0.05) or comparable quality compared to all other methods, except for two targets where MSAsubsample and SPEACH_AF perform better, and one target where AFcluster outperforms AFsample2. Specifically, compared to AFvanilla, MSAsubsample, AFcluster, SPEACH_AF, and AFsample, 9, 5, 17, 4, and 8 out of 23 targets are improved, respectively.

The performance of AFsample2 is significantly higher at the level indicated by the *P* value using a Wilcoxon-signed-rank test, see Fig. 3d. In addition, by analyzing the two cases where MSAsubsample performs better, P00558 and O76728 (see Fig. S2), it can be concluded that these cases would also have been improved with AFsample2 if a larger fraction of MSA masking had been used, see Fig. 2b. For both cases, 50% MSA masking, instead of the 15% used, would have yielded similar or better results compared to MSAsubsample. While these two cases are exceptions, they highlight the importance of exploring different levels of MSA masking to achieve optimal diversity.

By analyzing the difference between the TM-score for the best open and closed models for AFsample2 to AFvanilla, see Fig. 3e, it is evident that AFsample2 generates better conformations of the open state for most targets without significantly compromising the quality of the best-closed conformations. In only one case, the closed state of A0A075Q0W3 does the TM-score decrease by more than 0.05 units. As with the previously discussed examples, this case would also have benefited from optimizing the fraction of MSA masking used (see Fig. 2b). In this case, a lower fraction of MSA masking (0% or 5%) would have been ideal. This highlights the challenge that, while a lower masking fraction is sometimes optimal, a higher fraction may be required in other situations. Ideally, exploring different masking levels is possible but a complicating factor is that model confidences from different MSA masking levels are not directly comparable.

Finally, Fig. 3f illustrates three successful AFsample2 predictions of both the open and closed conformation. In these examples, there is a substantial difference between the open and closed states, and AFsample2 generates high-quality models for both conformations. In contrast, AFvanilla is only able to produce the closed conformation and fails to generate the open conformation for any of these cases.

## AFsample2 generates diverse protein ensembles

The effectiveness of AFsample2 in generating open and closed conformations has been presented above. However, the analysis only considered the best open and closed models generated, completely ignoring the ensemble of models between the two states. This ensemble of models might capture relevant micro-states that are on-path between the two states, or they might simply be unsuccessful attempts to model the different states. In order to capture the extent of conformational diversity, each model in the generated ensemble was compared with reference open and closed states using TM-align and visualized on a scatter plot that shows the similarities of the generated model ensemble to the two reference states, we choose to call this visualization *diversity plot*. Illustrated in Fig. 4a is the diversity plot for target P31133, where it is evident that all methods successfully generate the closed conformation (top left region in the diversity plot). However, only AFsample2, SPEACH_AF, and to some extent, AFcluster are able to generate both states, as well as several prospective states between the two open and closed conformations. MSAsubsample seems to tend towards the alternate state but fails, AFsample, on the other hand, successfully generates both states but has no models on the path between the states. Diversity plots for all targets in the OC23 dataset are compiled in Fig. S3. Overall, AFsample2 is able to generate both states and models between the states for a majority (15/23) of the targets in OC23

Although the presented example and visual cues from diversity plots (Figs. 4a and S3) indicate the effectiveness of AFsample2 in generating diverse models, a measure that captures the extent of diversity is required for a valid comparison. To this end, we developed the *fill ratio*. In short, the *fill*

*ratio* breaks down the path from open to closed state into 100 bins and calculates the percentage of bins that are populated by at least one model, exemplified for the diversity plots in Fig. 4a (see "Methods" for details). AFsample2 MSAsubsample and SPEACH_AF were observed to have a similar distribution of diversity, as quantified by *fill ratio*, with medians that are clearly higher compared to AFvanilla, AFcluster, and AFsample (Fig. 4b). This is good news since all three methods seem to have succeeded in their aim to introduce more diversity. Comparing the fill ratios for each method methods, target-by-target, shows that indeed AFsample2 has significantly higher fill ratios compared to AFvanilla, AFcluster, and AFsample ($P < 0.05$, Fig. 4b), while the improvement compared to MSAsubsample and is SPEACH_AF insignificant.

## Mapping intermediate states to the PDB

The validity of presumed intermediate states in ensembles of models generated by AFsample2 was assessed by searching in the PDB for intermediate structures with similar sequences (sequence similarity > 90%) to the target sequences. For four targets, P31133, Q7DAU8, P71447, and P33284, structural intermediates were found in the PDB. When mapped on the diversity plots for the AFsample2 ensembles, a high similarity (TM > 0.9) between the intermediate structure and model in the AFsample2 ensemble was observed for several cases, see Fig. 5 (top). As a reference, four examples where no structural intermediates and only structures referring to the end states were found are shown in Fig. 5 (bottom). This demonstrates that in some cases the models generated by AFsample2 are indeed likely to be true intermediate states.

## Fluctuations in predicted protein models agree with experimental data

As discussed earlier, AFsample2 induces randomness in the AF2 system in an attempt to diversify model predictions. However, it is not given that the diversity in the ensemble of models would actually agree with the experimentally observed fluctuations. To investigate this relationship, the Root Mean Squared Fluctuation (RMSF) of C$\alpha$ coordinates for generated model ensembles superimposed on the average model as determined by Pcons[20], was calculated for AFvanilla and AFsample2. The RMSF profiles were compared to the actual per-residue distance between the opened and closed states ($\Delta$C$\alpha$), exemplified in Fig. 6a for target B3EYN2 with the corresponding scatter plots for AFvanilla and AFsample2 in Fig. 6b, c. The RMSF profile for both AFsample2 and AFvanilla agree very well with $\Delta$C$\alpha$ ($R = 0.82$ and $R = 0.81$). In general, the correlations are slightly higher for AFsample2 compared to AFvanilla, see Fig. 6d. However, more importantly, AFsample2 seems to capture the magnitude of the fluctuations better to accommodate larger displacements, as illustrated by the example and in Fig. 6e. Here, the RMSF for different thresholds of $\Delta$C$\alpha$ is shown for AFvanilla and AFsample2. Overall, RMSF for AFsample2 is higher compared to AFvanilla for the same threshold, and for regions with larger $\Delta$C$\alpha$, the RMSF for AFsample2 is increasing faster compared to AFvanilla. This emphasizes that the conformational diversity in AFsample2 ensembles is enhanced compared to the AFvanilla ensembles. Per-residue fluctuations, scatter plots, and the correlation coefficients between the RMSF profile and $\Delta$C$\alpha$ for all targets in the OC23 dataset are summarized in Fig. S4.

## Selecting protein states

Given an ensemble of generated models, it is non-trivial to correctly identify conformational states without the availability of reference structures. Principle component analysis is commonly used to visualize variation in model ensembles[13]. However, it does not provide a way to select which models to represent the different states, see Figs. S5 and S6. Alternatively, structural clustering can be used to get representative states, but it is problematic to automate clustering since the clustering thresholds will depend on the structural difference among the models in the ensemble, which is not known a priori. Instead, AFsample2 introduces a simple strategy to identify two conformational states given an ensemble, the method relies on the tendency of the AF2 inference system to favor a particular state by default.

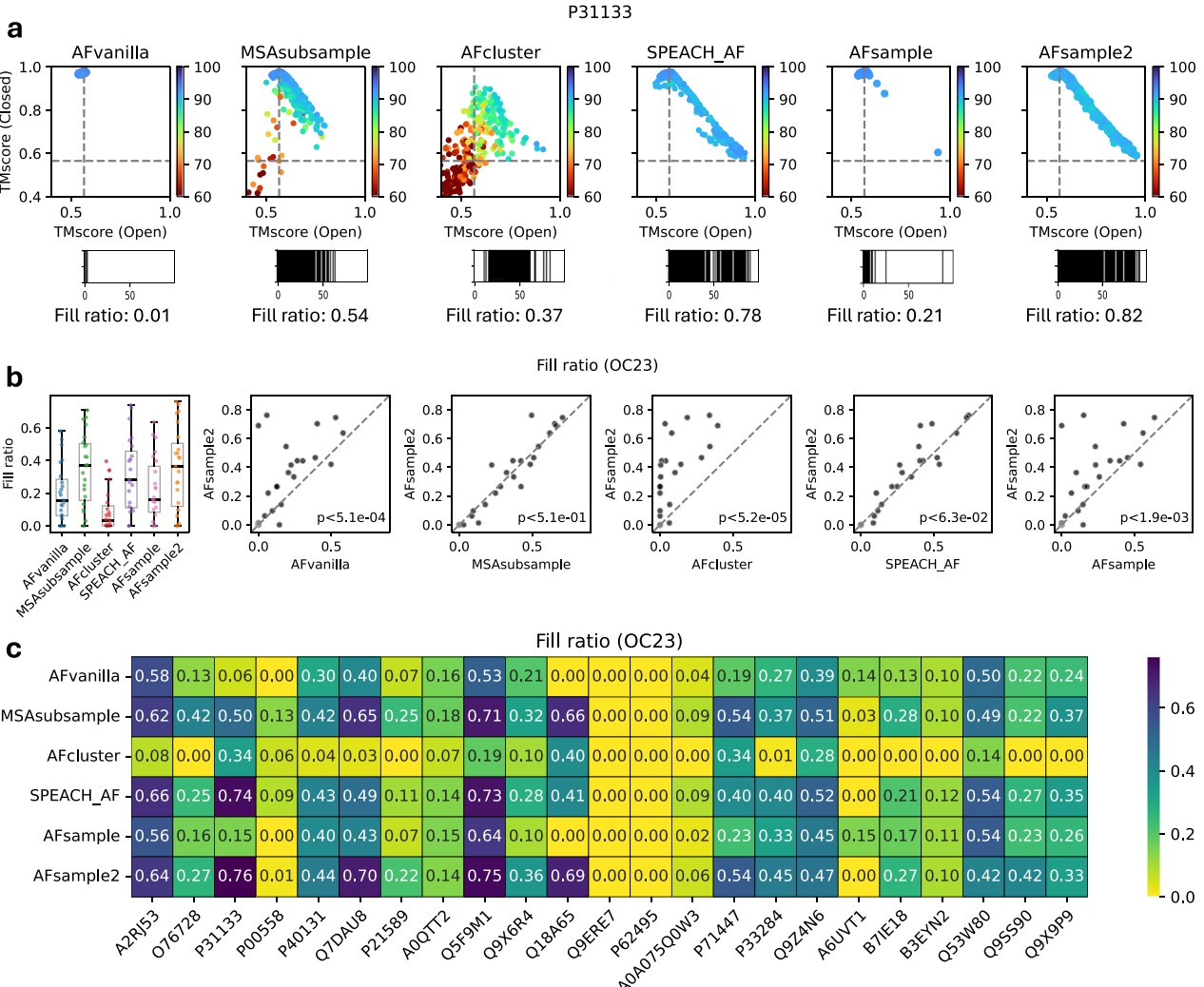

**Fig. 4 | Analyzing the diversity of generated ensembles. a** Diversity plot showing similarity of generated models with open and closed reference structures for AFsample2 ensemble on selected target from OC23 dataset along with an indicative representation of fill ratio. **b** Comparing *fill ratios* of the model ensembles generated by all methods under consideration for all targets in the OC23 dataset. **c** Heatmap summarizing *fill ratios* for all targets in the OC23 dataset when compared between methods.

Assuming the AF2 inference system ranks one state higher in terms of confidence (best model) and that the AFsample2 system promotes diversity, models in the alternative state are expected to differ significantly from the best model in terms of structural similarity (TM-score). Consequently, the alternative state should appear as a high-confidence model that is structurally distinct from the best model.

In short, this strategy follows a three-step process, starting with (i) calculating the similarity to the best model for all models in the ensemble, (ii) *Confidence screening*—for filtering models below a certain threshold, and (iii) *Extremity selection*—to identifying the model that is furthest from the most confident model. A visual representation of the idea of selected high-scoring models structurally dissimilar from the best model is depicted in Fig. 7a. This strategy works best for two-state systems but could potentially be extended to more states by iterating and calculating the maximal similarity to selected states.

This approach to identify the alternate state was benchmarked on model ensembles generated by both AFsample2 and AFcluster individually to ascertain the robustness of the method across varied ensemble characteristics. The quality of the selected states was assessed using the mean squared error (MSE) between the TM-scores of the selected states and the best possible selection. The best possible selections are the two models from the generated ensemble that are most similar to the two reference states. This

is an upper bound on the quality of the selected models. The *confidence screening*, step (ii), requires a threshold, and we monitored the MSE as a function of this threshold for AFsample2, and AFcluster, Fig. 7b, e. We choose to make the threshold relative to the highest confidence rather than absolute to account for predictions of varying quality. For AFsample2 there is actually almost no need for a threshold, as the selection is equally good even without any threshold (0% of highest). For AFcluster, on the other hand, a threshold is needed since those ensembles contain many low-confident models, exemplified in Fig. 7j. Around 90% and 85% of the highest confidence seems to be the optimal threshold for AFcluster and AFsample2, respectively. But ideally, it is best to manually inspect the reference-free plots Fig. 7i, k to locate the alternative states. For the optimal threshold, the MAE is around 0.06 TM-score units for both AFsample2 and AFcluster, however, the MAEs are not comparable since the ensembles are different.

The actual similarity of the selected states and the best possible selection is summarized in Fig. 7c, d, f, g for AFsample2 and AFcluster, respectively. In almost all cases, the selection is relatively close to the optimal selection. AFsample2 has one poor selection (MSE > 0.27) of the closed state for target B7IE18, the selected model is 0.55, while the best is 0.92. The selected models by AFcluster have no such failures, but since the AFcluster ensembles are of lower quality, the risk of failure is also lower. Reference-free plots for all targets in the OC23 dataset are shown in Fig. S9.

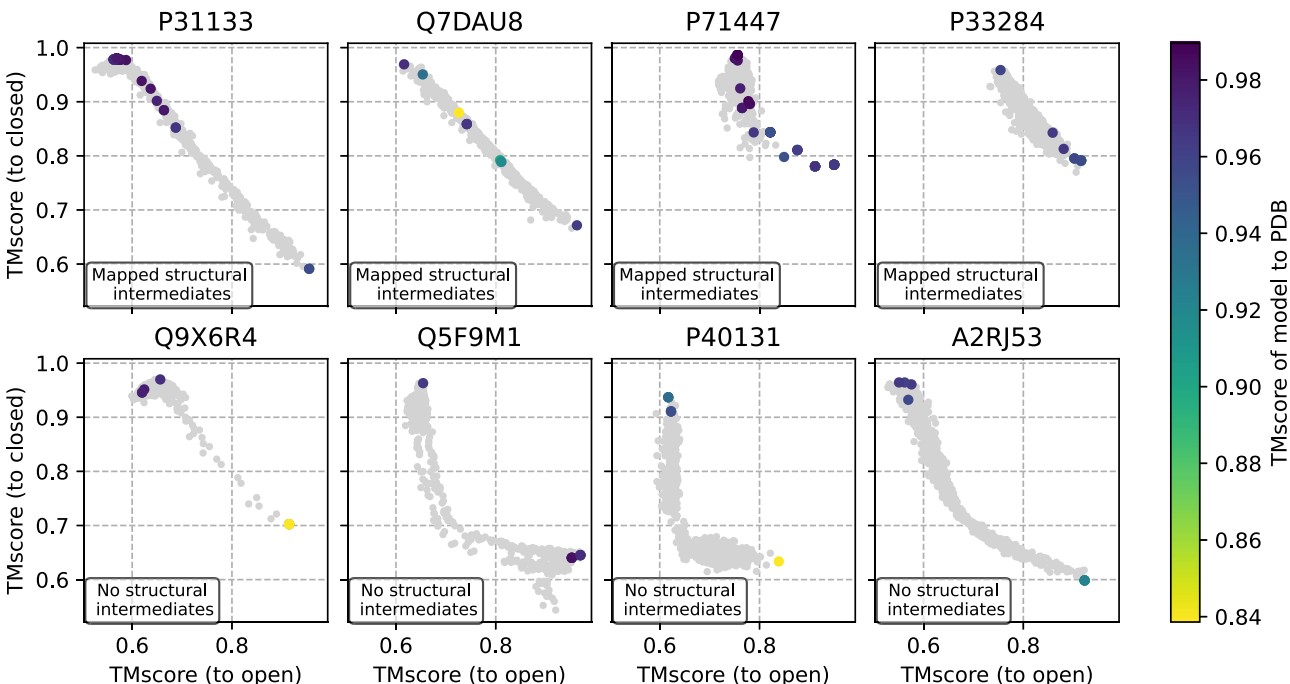

**Fig. 5 | AFsample2 ensembles mapped to PDB.** Experimental structures in the PDB with similar sequences (>90%) are shown with their corresponding TM-score. The closest model mapped to the PDB (sequence similarity > 90%) is annotated with colors. Top row show cases where it was possible to find structural intermediates, and bottom row show cases where no intermediates were found in PDB.

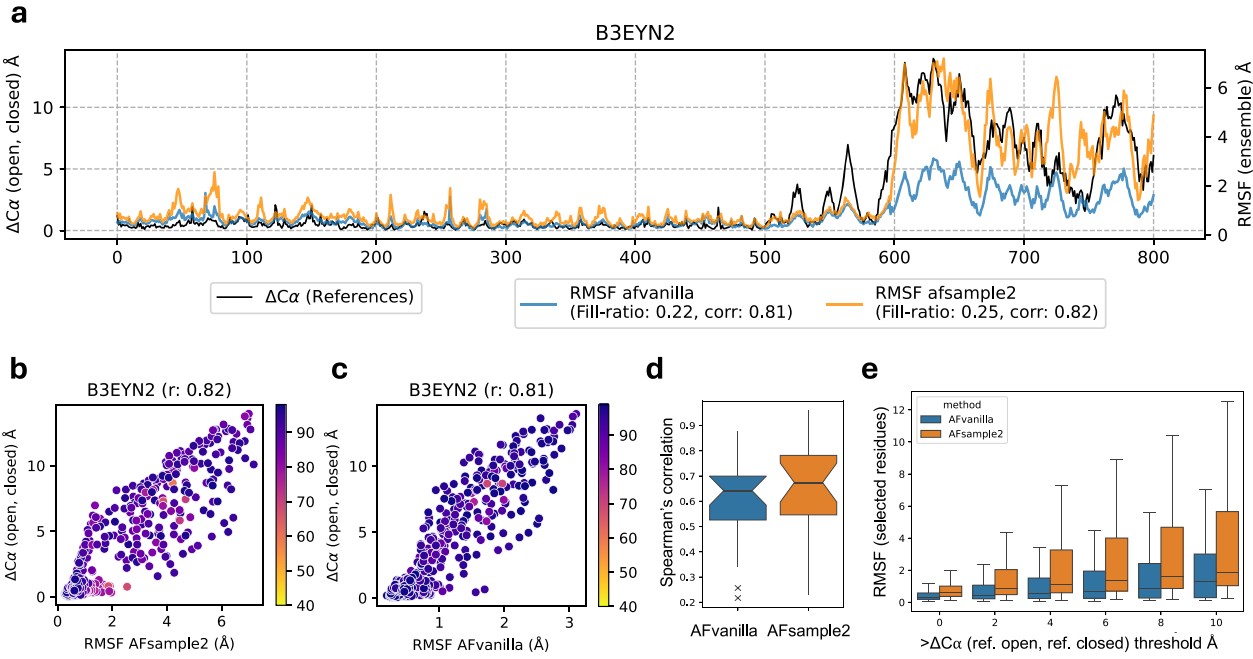

**Fig. 6 | Comparing ensemble fluctuations with reference states. a** Analyzing the amount of residue fluctuations observed by all methods in consideration. **a** A consolidated analysis of per-residue fluctuations observed in the model ensemble for B3EYN2 and its correlation with C-α distances between experimental states, for AFvanilla and AFsample2 (**b**, **c**) Comparing per-residue correlation for B3EYN2 between AFsample2 and AFvanilla. **d** Aggregate per-residue correlation for all targets in the OC23 dataset. **e** Comparing RMSF between methods at multiple distance thresholds.

The selection process is illustrated for target Q5F9M1 in Fig. 7h–k for AFsample2 and AFcluster, respectively. The selected states are indicated in the reference-free plots by a red arrow. The gap of ~0.1 TM units between the optimal selection and the rest of the model ensembles arises because the target sequence is taken from UniProt, and includes flexible termini that are consistently dissimilar to any other model in the ensemble. Despite the difference in the model ensembles between AFsample2 and AFcluster, the proposed method to identify the alternate state is able to select relatively high-quality models of both states irrespective of the method used for generating the models. This is true for most targets in the OC23 dataset, see Fig. 7l.

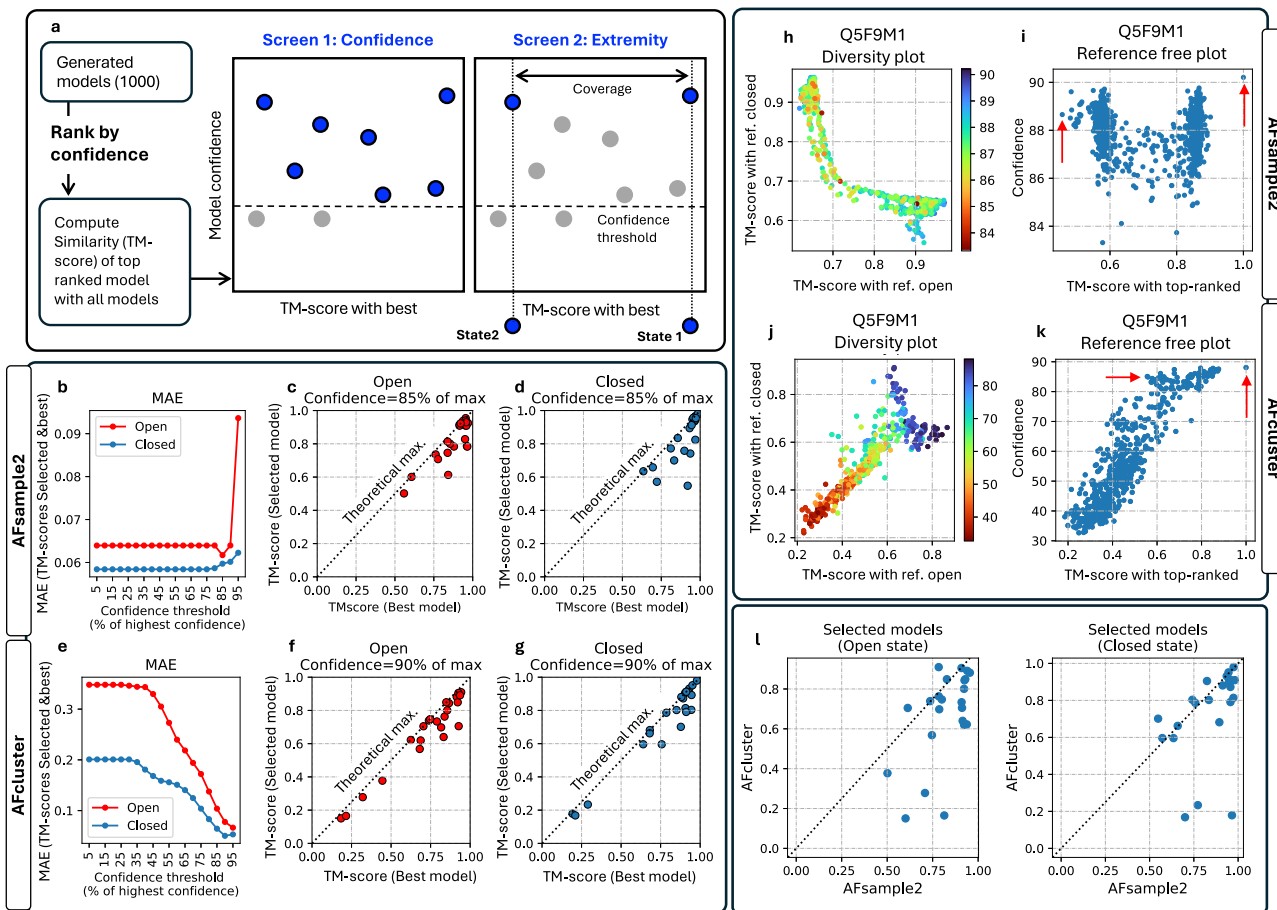

**Fig. 7 | Reference-free state determination. a** Schematic illustrating the process of identifying open and closed states from a pool of generated models. **b, e** Mean absolute error (MAE) between the TM-scores of selected and best possible model for AFsample2 and AFcluster. **c, d** Scatter plot between TM-scores of selected models and best possible models generated by AFsample2 for all targets in the OC23 dataset (**f, g**—for AFcluster). An example with target Q5F9M1 showing (**h**) diversity plot and **i** reference-free plot. The reference-free plot has been annotated with optimal selections for the open and closed state (**j, k**—AFcluster). **l** Comparing selected open (left) and closed (right) models with the reference-free strategy for AFsample and AFcluster.

## Evaluating AFsample2 on additional datasets

As described earlier, AFsample2 performs well in predicting both the open and closed state on the OC23 dataset. However, it is important to demonstrate the same level of effectiveness for additional datasets. To this end, a dataset of 16 transporter proteins, which contains the "inward-facing" and "outward-facing" conformational states, was utilized (see "Methods"). 1000 models were generated for each method and the result was analyzed. We first compared the best-generated models using AFvanilla, MSAsubsample, AFcluster, SPEACH_AF, AFsample, and AFsample2, using scatter plots and distributions, see Fig. 8a, b. Individual TM-scores for all targets and methods are shown in Fig. S2b. It should be noted that this dataset is fundamentally different from the OC23 as models generated by AFvanilla do not show a bias towards one of the conformations. This makes it difficult to achieve significance for the generation of inward- and outward-facing states since the improvement is shared between both states. In addition, the targets in the set are in general more difficult, i.e., the final model quality is lower. In fact, while all methods except AFcluster successfully predict one of the states with TM > 0.8, for all but one target, it is only AFsample2 that is able to predict both states for 50% of the cases (8/16), the best of the other methods predict at most 4 targets. This improvement is also illustrated in the AUC plot between the fraction of successful proteins, defined as having both states above a certain TM-score (TM-score > threshold), see Fig. 8c. Here, AFsample2 has the highest AUC, followed by MSAsubsample and SPEA-CH_AF. Furthermore, we also compared the generation of models for each method against AFsample2 using the minimum TM-score for the two states

as a measure of performance (Fig. 8d). In almost all cases, models generated by AFsample2 demonstrate higher (ΔTM > 0.05) or comparable quality compared to all other methods, except for two targets where MSAsubsample perform better, and one target where AFcluster and SPEACH_AF outperforms AFsample2. Specifically, compared to AFvanilla, MSAsubsample, AFcluster, SPEACH_AF, and AFsample, 11, 7, 12, 7, and 9 out of 16 targets are improved, respectively. The performance of AFsample2 is significantly higher compared to MSAsubsample and SPEACH_AF ($P < 0.05$) and to AFvanilla, AFcluster, and AFsample ($P < 0.001$), using Wilcoxon-signed-rank test, specific $p$ values in Fig. 8d.

## Case study: modeling difficult fold-switches with AFsample2

The inability of AF2, even with MSA subsampling, MSA clustering, and shallow MSA, to model intrinsically disordered proteins/regions (IDPs) and fold-switch proteins has been reported[21,22]. This limitation has largely been attributed to the lack of IDPs in the training set of AF2. Although methods like AFcluster were originally developed and tested on fold-switch proteins, the results in a recent study indicate the ineffectiveness of *all* MSA sampling methods in modeling a modified S6-ribosomal protein that alters conformations between an $\alpha/\beta$-plait (FS1) and a $3\alpha$-helical (FS2) fold[23,24] dependent on temperature, see Fig. 9.

We generated 1000 models with AFsample2 and AFcluster for the modified S6-ribosomal protein, indeed the majority of the generated high-confident models are similar to the FS1 state (TM-score > 0.8), this is true for both AFsample2 and AFcluster. However, for AFsample2, there is a handful

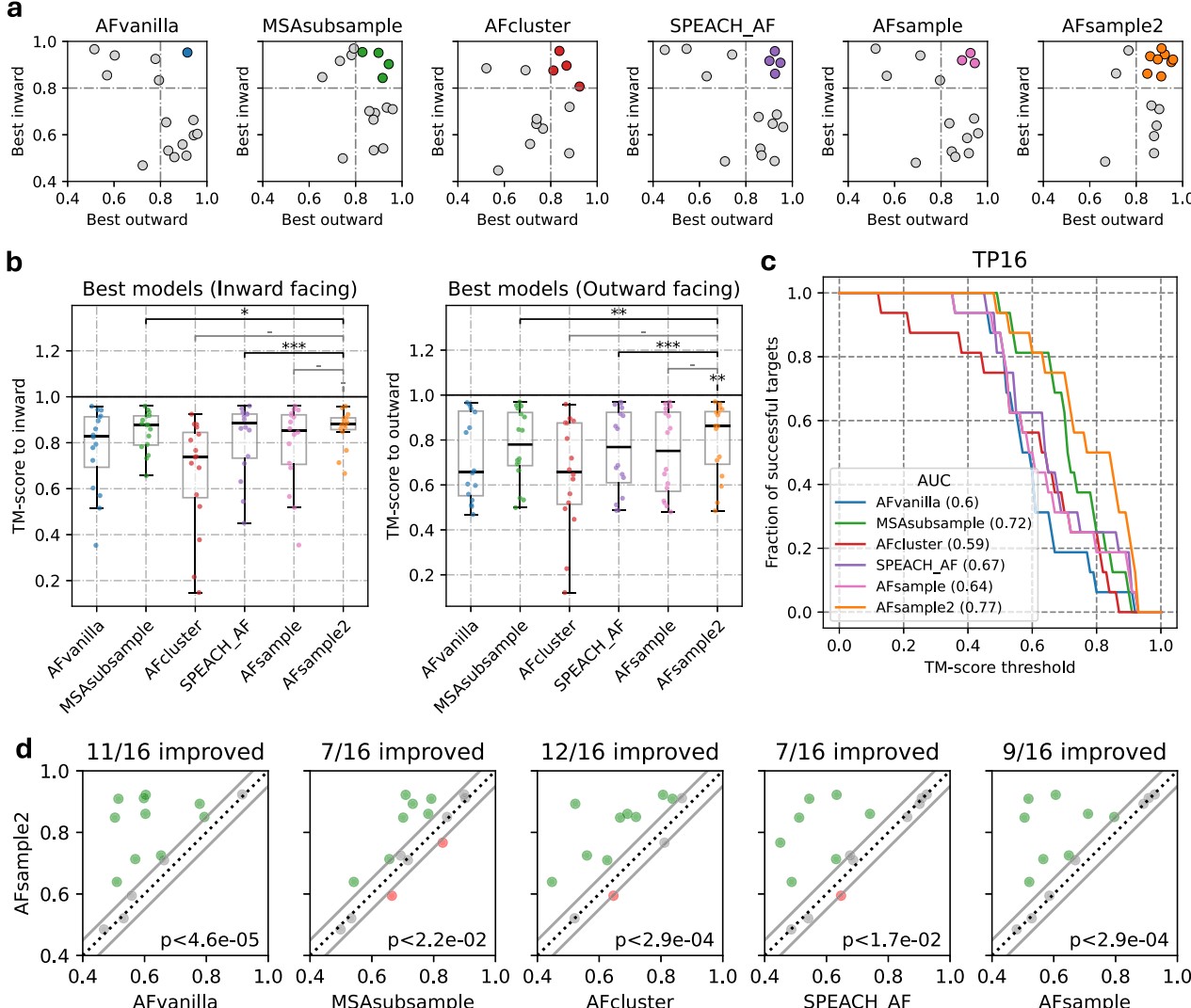

**Fig. 8 | AFsample2 tested on a transporter dataset with 16 targets. a** Comparing TM-scores of best models generated for inward-facing and outward-facing states for all targets and methods on the TP16 set. **b** Distribution of best-generated models for the TP16 dataset by all methods under consideration in terms of similarity (TM-score) to the reference states show significant (Wilcoxon-signed-rank test *p* value: \*<0.05, \*\*<0.01, \*\*\*<0.001, − ns, *n* = 16). **c** Fraction of successful targets compared among methods at different levels of TM-score thresholds. Success: TM-score > threshold (for both states). **d** Comparing the minimum of the best selection for all methods against AFsample2 *ΔTM* < 0.05 in gray, green AFsample2 improved, red AFsample2 worse.

of models that are highly similar (TM-score > 0.8) to the FS2 state, see Fig. 9. These models have lower model confidence compared to the FS1 state (0.75 vs 0.88), but from the reference-free plot (Fig. 9f), they can still be identified as a relatively confident alternate state. Generating more models could potentially yield higher-scoring models for the alternate state.

Current efforts that use AF2 for modeling alternative states of disordered proteins rely on model confidence to select models. However, as AF2 was trained to estimate high-quality models for a single conformation, the confidence metric might not be a reliable indicator while looking for alternative states as the AF2 inference system is bound to rank these structures lower. This was also observed in the OC23 set, where the open conformation was consistently less favorable in terms of model confidence Fig. S3. Instead, above a minimum confidence score, the similarity to the highest confidence state would serve as a better indicator of model diversity.

In summary, current methods that aim to improve conformational diversity tend to be sub-optimal for fold-switch proteins. AFsample2 shows that it is possible to utilize the AF2 inference to make predictions for proteins that have not been part of its training set and that AF2 can generalize to new unseen proteins.

## Discussion

Sampling and clever MSA reshuffling have been shown to improve the conformational coverage of models generated using the AF2 inference system. However, solutions are required to estimate better (i) end states as well as (ii) intermediate states to understand the full picture of the conformational dynamics. Additionally, there are many unsolved questions, such as how to correctly identify states without a priori knowing the structure of the state and how the states are physically and dynamically connected. In addition, we are struggling with insufficient experimental data where, in many cases, the structure of only one state is known, usually the most stable under experimental conditions, and we cannot tell with certainty if our predictions are correct. Of course, this is also an opportunity for computational methods to provide experimental hypotheses that can be used to validate the predictions.

A major strength of the AF2 inference system is its efficient extraction of the co-evolutionary structure embedded in MSAs. This capability is highly beneficial for generating a single, high-confidence model. However, when attempting to expand the conformational landscape of predictions, this specific feature of the AF2 system becomes a drawback, as the strong co-evolutionary constraints prevent the generation of alternative states.

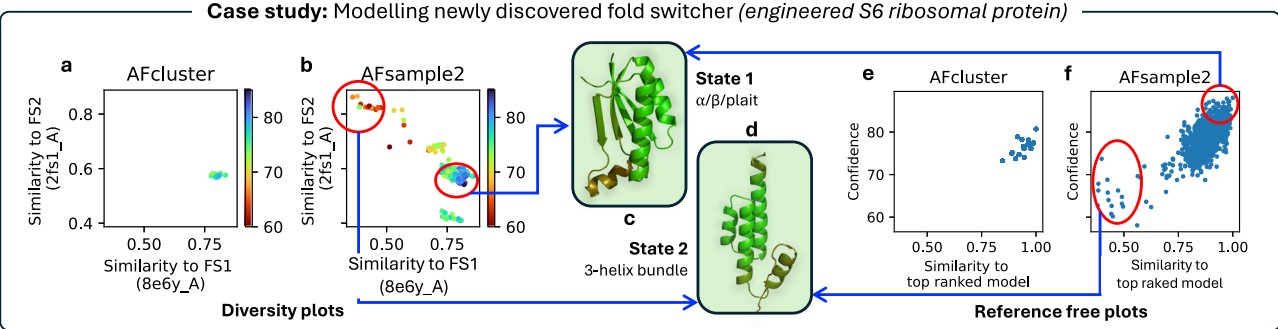

**Fig. 9 | Modeling a fold switcher with example of a S6-ribosomal protein.** **a, b** Diversity plot showing the similarity of the generated ensemble to both folds. The models associated with FS1 and FS2 are annotated with (**c, d**) structures of both folds. Furthermore, the alternate fold can also be visualized on the scatter plot generated with the (**e, f**) reference-free state determination protocol for the AFsample2 ensemble.

As a solution to these problems, we developed AFsample2, a method to generate high-quality conformational ensembles for proteins by randomized MSA column masking. We introduce noise within the MSA feature profile that leads to a significant improvement in the estimation of alternative states as demonstrated on diverse sets of proteins. Under the assumption that MSAs contain information about the patterns of correlated mutations (co-evolution) between residues that can inform physical contacts within the protein structure, making changes to these relationships is a viable strategy for inducing diversity in predictions. The idea is not limited to the AF2 inference system but could be readily applied to other MSA-based prediction systems, such as RosettaFold[9], OpenFold[25], or AF3[26].

In addition, we used AFsample2 here to predict different states, but it could also be utilized to improve the conformational sampling of difficult single-state proteins that seem to be stuck in one local minimum. In terms of limitations and future directions, the current version of AFsample2 has only been tested on monomers. However, it can be easily adapted for generating conformational states for multimeric protein complexes as well.

## Methods
### AlphaFold version
The entire prediction pipeline is based on AlphaFold v2.3.1, which is available at https://github.com/google-deepmind/alphafold. The original transformer model assumes that the embedding vectors do not correlate with each other[27]. However, in the case of the AF2's Evoformer, the relationships between MSA rows (alignments) and columns (residues) were established with row-wise and column-wise attention, respectively[8]. Randomized MSA masking is aimed at diluting these co-evolutionary signatures and increasing the probability of sampling alternative states.

### Datasets
Below, we describe the different datasets used in this study.

**OC23.** A set of 23 proteins with distinct open and closed states was selected as defined by a TM-score difference between the open and closed state <0.85 (Table S1). Protein sequences extracted from the experimental open and closed structures could have been directly used to run AFsample2 predictions. However, due to irregularities such as (i) Unequal chain length, (ii) missing residues, and (iii) incomplete information, we chose to retrieve the full protein sequence from Uniprot. With this, we were confident that the inference system was getting the complete information regarding the protein's sequence. In summary, all predictions were performed on the full protein sequence from UniProt for each protein.

**OC > 85.** A set of 22 proteins with smaller differences between open and closed states, TM ≥ 0.85, was assembled to analyze the effect of induced diversity on targets with small or no conformational diversity (Table S2).

**Transporters.** For the validation sets, a transporter dataset with 16 unique proteins with inward- and outward-facing conformation was utilized (Table S3). This dataset was manually curated in a recent study to benchmark protein structure prediction methods for membrane proteins that show substantial conformational changes[28].

### MSA generation and randomization
The MSAs were produced using the DataPipline implemented in AF2 with default parameters using HHblits[29] and Jackhammer searches on Uniref90[30], BFD, Uniclust30[31], and MGnify sequence databases. The MSAs are available for download. Templates were not used in any of the runs. Randomized MSA masking was incorporated into the pipeline at the MSA pre-processing step. The algorithm sequentially initializes model runners associated with specific model names on a given FASTA sequence. The user-defined parameter *–msa_rand_fraction* determines what fraction of MSA columns that should be randomly replaced with "X", ignoring the first row, which is the target sequence.

### Inference
Since multiple versions of trained model weights are available for AF2, the choice of the weights was an important consideration. AF2 has two versions of the monomer neural network weights, with each version having an additional five sets of neural network weights. The first version used during CASP14 was extensively validated for structure prediction quality[8]. The additional five pTM models were fine-tuned to produce pTM (predicted TM-score) and (PAE) predicted aligned error values alongside their structure predictions. The relative difference in these models was assessed in terms of the TM-score profile of the predictions with experimental open and closed states. All ten model-types were tested individually to ascertain if any of them was consistently doing better than the others. The results summarized in Fig. S8a, show that every model-type has the ability to generate the best model depending on the protein. Also, Fig. S8b indicates that the quality of models in terms of confidence did not vary much with different model weights. Based on this, and to achieve maximum performance, the AFsample2 pipeline utilizes all ten neural network weights for inference.

### AFcluster
AFcluster was run with the default parameters as described in their GitHub repository (https://github.com/HWaymentSteele/AF_Cluster) with the following command: `python ClusterMSA.py EX -i msa.a3m -o msas`.

### MSAsubsample
MSA subsampling was implemented in AFsample2 by modifying *max_extra_msa* and *max_msa_clusters* before running AlphaFold. Based on the method described in[12], *max_extra_msa* was used to randomly set the depth to any of these values: [16, 32, 64, 128, 256, 512, 1024, 5120] and

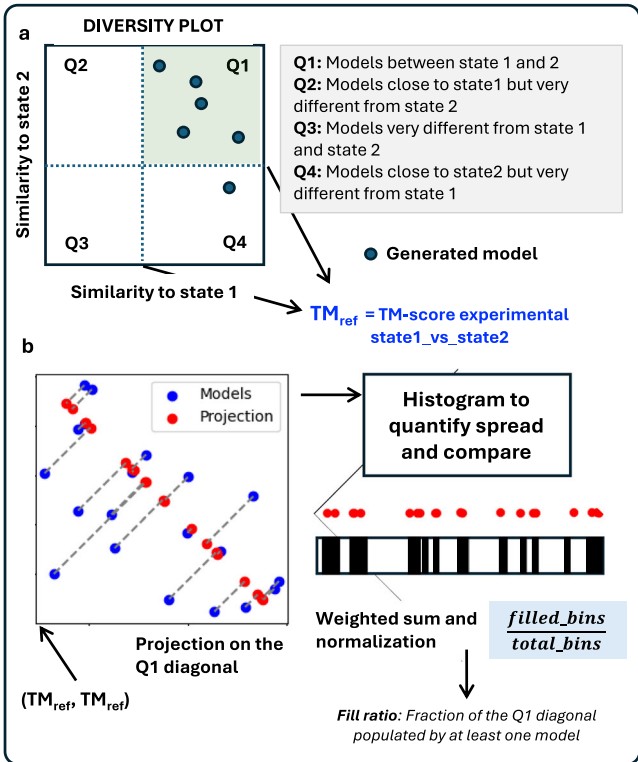

**Fig. 10 | The fill-ratio metric.** Given **a** diversity plot, this protocol quantifies the conformational diversity of ensemble models. **b** All models in Q1 are projected onto the diagonal. Models in other quadrants (Q2, Q3, and Q4) are ignored since they are not close to any of the two reference states. The diagonal is then divided into 100 bins, and the ratio of bins containing at least one model is recorded. This ratio gives information about the conformational space spanned by the model ensemble between the two reference states.

*max_msa_clusters* was set to half the depth or maximum 512. In total, 1000 models were generated per target.

### SPEECH_AF

The positions in the MSAs to change to alanine were computed with the code provided here: https://github.com/RSvan/SPEECH_AF. Since the outputted MSAs did not run with the current version of AlphaFold, the actual modifications of the MSA were implemented in AFsample2. Thus, there is no need to generate multiple MSAs; instead, a list of positions to change is inputted to AFsample2, and the original MSA is modified accordingly at runtime.

### AFsample

AFsample was run with three recycles, and dropout was turned on at inference using the code available here: https://wallnerlab.org/AFsample, increased recycles, and templates were not used.

### Evaluation metrics

**TM-score**. TM-score[32] calculated using TM-align[33] was used as the evaluation metric for comparing similarity to reference states and to other models. TM-score was calculated using a fixed $d_0 = 3.5$Å to avoid the problem of long proteins achieving artificially high TM-score[32].

**Fill-ratio**. To quantify conformational diversity given an ensemble of models, the fill-ratio metric was developed. The fill-ratio metric involves projecting points, representing models in an ensemble, in the diversity plot onto a diagonal defined by the equation $y = mx + q$, where $m = -1$

and the intercept $q$ is calculated based on the starting x-coordinate $TM_{ref}$ (TM-score between experimental states), see Fig. 10a. For each point $(x, y)$, its projection onto the diagonal is calculated using the following steps.

$$x_p = \frac{x + my - mq}{m^2 + 1}, \quad y_p = m \cdot x_p + q$$

where $(x_p, y_p)$ represents the projected coordinates of the point $(x, y)$ onto the diagonal, see Fig. 10b.

Next, the scalar position $s$ of each projected point along the diagonal is calculated as the Euclidean distance from the starting point $(TM_{ref}, 1)$ of the diagonal:

$$s = \sqrt{(x_p - TM_{ref})^2 + (y_p - 1)^2}$$

This scalar position $s$ represents the distance along the diagonal, from the starting point, to each projected point.

The total length $L$ of the diagonal is calculated using the Euclidean distance between the starting point $(t_{ref}, 1)$ and the ending point $(1, t_{ref})$:

$$L = \sqrt{(1 - TM_{ref})^2 + (1 - TM_{ref})^2}$$

The diagonal is then divided into $N$ equal bins, where $N = 100$, and the width of each bin is given as bin width = L/N Each scalar position $s$ is mapped to a bin index $k$ as follows:

$$k = \left\lfloor \frac{s}{Bin\ Width} \right\rfloor$$

The floor function ensures that each scalar position $s$ is assigned to the correct bin, and the bin indices are clipped to ensure they lie within the range $[0, N-1]$. Furthermore, to emphasize that it is important to fill the bins near both ends of the diagonal, we applied a parabolic weighting function.

$$w(k) = 1 + 16 \cdot \left( \frac{k}{bins - 1} - 0.5 \right)^2$$

This ensures that generating alternate conformation remains the highest priority. Finally, weighted ratio of populated bins is computed as follows to get fill ratio for the given ensemble.

$$Fill-ratio = \frac{\sum_{k \in populated\ bins} w(k)}{\sum_{k=0}^{bins-1} w(k)}$$

The method outputs both the ratio of populated bins and the indices of the populated bins. This can be applied in the analysis of protein structures to (i) Analyze the spread of models in the TM-open vs TM-closed plot, (ii) Quantify the degree of overlap or clustering of data points, and assess the quality and completeness of data representation.

### Reporting summary

Further information on research design is available in the Nature Portfolio Reporting Summary linked to this article.

### Statistics and reproducibility

All statistical analyses were performed using Python together with the NumPy and SciPy libraries. Wilcoxon-signed-rank tests were used to determine whether the performance difference for a given method was significantly higher than that of AFsample2. The sample sizes were 23 for OC23 and 16 for TP16.

**Article**

All data and code have been made available on Zenodo and GitHub to ensure the reproducibility of the experiments and the presented results. In addition, a separate Jupyter Notebook has been provided to exactly reproduce the analysis and all figures in this study.

## Data availability

All data presented in this study are made available at Zenodo[34], along with the *reproduce_figures.ipynb* notebook to reproduce all figures.

## Code availability

All scripts presented in this study are made available at https://wallnerlab.org/AFsample2.

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

## Acknowledgements

This work was supported by the Wallenberg AI, Autonomous System and Software Program (WASP) from Knut and Alice Wallenberg Foundation (KAW), Swedish Research Council, grant 2020-03352, The Swedish e-Science Research Center, and the Wenner-Gren Foundation, grant UPD2023-0023. The computations were performed on resources provided by KAW and NSC (Berzelius).

## Author contributions

Conceptualization: B.W.; methodology: Y.K., B.W.; software: Y.K.; investigation: Y.K., B.W.; data curation: Y.K., B.W.; writing—original draft: Y.K., B.W.; writing—review and editing: Y.K., B.W.; funding acquisition: B.W.

## Funding

## Competing interests

The authors declare no competing interests.
