## [Transparent Peer Review file · Communications Biology]

AFsample2 predicts multiple conformations and ensembles with AlphaFold2

Corresponding Author: Professor Bjorn Wallner

Version 0:

Reviewer comments:

Reviewer #1

(Remarks to the Author)

This manuscript describes another method, AFSample2, aimed at increasing model diversity with AlphaFold2. It compares results from the new method to three other methods that have been proposed to also increase diversity and the base AlphaFold2 method. In addition, they propose a new method to identify alternate protein conformations. This method is a derivation of the previously published method, SPEACH_AF. While the results presented are of interest, it is unclear why they propose a new method for comparing models without using ground truth structures when principal component analysis (PCA) is a widely used method to examine conformational diversity in the absence of ground truth structures. In addition, I feel the improvement over AFvanilla is overstated.

1. The manuscript should clarify the exact number of cases where AFSample2 improves upon AFvanilla:
 - 1a. Of the 17 cases in the OC23 set where AFSample2 improves the TMScore of the open conformation, 7 of these already yield both conformations with AFvanilla based on the TM-score>0.8 criteria and 3 of these do not generate models that cross the 0.8 threshold, yielding 7 cases of having both an improvement over AFvanilla and yielding both conformations.
 - 1b. "On the other hand, AFsample2 is able to successfully generate high-quality models of both states without failing for even a single target in the OC23 dataset." There appear to be three cases, P00558, Q9ERE7, and P62495, where both conformations do not achieve a TM-score>0.8.
2. Concerns regarding the RMSF analysis and presentation.
 - 2a. The assumption is that the $\Delta C\alpha$ are the wings of the normal distribution for the disorder of the models, RMSF, spanning from open to close. The correspondence between the calculated slope/Z-score and the expected value needs further explanation, especially for cases where both conformations are not achieved, the model distribution is not normally distributed, or the fill factor is low.
 - 2b. The legend should clearly indicate the meaning of the red lines in Figs 5b-f and S4. If they are the fit lines, why do some of them not appear to match the data very well?
 - 2c. The reason for the split data in some of the plots should be discussed.
3. "Identify protein states without ground truths".
 - 3a. A comparison of this new method to PCA should be carried out.
 - 3b. The reference structures used to determine the best possible selection should be clarified. Are they a crystal structure?
 - 3c. The large gap between the optimal selection and the rest of the models in Figs. 6i, k and S8 needs explanation.
 - 3d. The method's ability to return the open and closed states should be evaluated and discussed.
4. In the comparison with the Transporters set, the threshold for indicating AFSample2's success was set at 0.6, which is lower than the 0.8 threshold used for the OC23 set. This lower threshold should be explained. Additionally, the rationale for showing AFcluster in Fig. 7g should be provided, since in the original Xie and Huang paper, MSAsubsampling was reported to be better than AFcluster at obtaining both conformations.
5. The comparison of AFSample2 to SPEACH_AF on the same set of proteins should be carried out.
6. The requested additional method details should be included in the manuscript.
 - 6a. What were the parameters used for MSAsubsampling.
 - 6b. What were the parameters used for AFcluster.
 - 6c. Was dropout used for AFSample2? If so, wouldn't dropout potentially reduce the effect of column masking?
7. The legends for Figures 3 and 5 are incomplete and should be updated.

Reviewer #2

(Remarks to the Author)

Kalakoti and Wallner describe in this study a new approach named AFsample2 for predicting conformational ensembles with AF2. AFsample2 is benchmarked against many different previously reported strategies, and the results show a significant improvement when predicting additional states. Of relevance is the fact that AFsample2 was shown to work also with a case example of a IDP, which is interesting providing the inability of the other strategies. The paper is interesting, however, it should be much better stated and probably visually represented in the paper how different is Afsample2 with respect to the previous approaches. I understand that AFsample2 randomly masks MSA positions to Ala, but this strategy was also applied in previous studies (for instance SPEACH_AF, and recently in 10.1021/acs.jctc.4c00222). It should also be better stated that AFsample2 also increases the sampling rate and introduces noise by enabling dropout layers at inference as AFsample. Therefore it should be much better stated what is unique of Afsample2 and how it differs to other approaches, otherwise it is hard to judge the novelty of the approach. In addition to that, the paper misses some contributions to the field, such as the recent 10.1021/acs.jctc.4c00222, or 10.1002/pro.4426 in which different templates coming from either X-ray or MD data were included. Some of the included bioarxiv references need to be updated as well.

I have also the following comments:

- In Figure 5, the RMSF for the different strategies is represented for one of the studied cases. RMSF coming from MD simulations is often similar to the experimental B-factors, although I agree there might be some deviations coming from the different crystallization and simulation conditions. Still, it is a good practice to compare RMSF with B-factors, and so I believe it would be nice to include the experimental B-factors and compare whether the flexibility trends are well reproduced.

- As shown in Figure 4 and later figures, AFsample2 can provide open, closed and intermediate states of a given protein. The TM-score of the best open and closed models are computed. However, I wonder about the physical validity of the intermediate states. I understand that only structures with high pLDDT scores are taken, but still, in my opinion it would be nice to generate a PC space with all available PDB structures of the studied protein, and plot on this experimental X-ray based PC space the multiple open, closed and intermediate structures generated with Afsample2.

Reviewer #3

(Remarks to the Author)

The author introduces AFsample2 for predicting multiple conformations and ensembles with AlphaFold2. The author introduces AFsample2 for predicting multiple conformations and ensembles with AlphaFold2. This issue is very important for studying protein mechanisms. Here are some questions I hope the author can address:

1. The author samples different protein conformations by randomly masking the MSA and provides what they believe to be appropriate parameters: 1000 samples at 15% masking. I would like to know whether the author has conducted repeated experiments? Is there randomness involved? Especially given that the test dataset is not very large, is this conclusion reliable?
2. The complete protein dataset and the author's experimental results were not accessible through the provided link. Could the author provide the relevant data, including the predicted structures, to facilitate reproduction of the results?
3. The datasets presented by the author, OC23 and Transporters, are relatively small, and the author might consider further expanding them. In addition, these test proteins seem to have only the final state structures. Is there an example that can provide a comparison between the real ensemble and the ensemble predicted by AFsample2?
4. The author provided a parameter of 15% masking in the paper. I would like to know the impact of random MSA masking on the performance of AF2 structure prediction.
5. AFsample2 predicts ensembles by random masking the input MSA. Previous similar works, such as changing the number of input MSAs, have shown that reducing the MSAs leads to a significant decline in the quality of the predicted structure. Could the author discuss whether modifying or reducing MSA information to predict ensembles is reasonable? After all, AF2 was originally designed to predict static structures, and ensembles were not part of the initial design framework.
6. Has the author considered altering template information to predict ensembles?
7. Do the predicted structures obtained by changing the input truly reflect the native structure? For example, are there cases where experiments do not show that the protein has multiple conformations, but the predicted structure is very different after changing the MSA?
8. How do authors determine whether a protein has multiple conformations? Or how to assess how many final state conformations a protein has?
9. Has the author studied ensembles of complexes? Are there relevant datasets and experimental results that can be shared? For example, has AFsample2 been further extended to AF2-multimer?

Version 1:

Reviewer comments:

Reviewer #1

(Remarks to the Author)

The authors have made significant improvements based on the initial reviews. The comparison to the other methods supports the utility of this methodology. Even if not entirely novel, the ability to yield more diverse conformations and additional intermediates for even one example is worth publishing. However, there are still a few points that should still be addressed:

1. The statement about SPEACH_AF relying on having experimental structures or predictable models for one of the states is true, but the authors should note that AFvanilla is able to generate one of the conformations in all cases, so this argument seems spurious as SPEACH_AF uses an unmodified MSA. It is also surprising that no score is reported for A6UVT1, as the unmodified MSA should yield a viable conformation.
2. Two of the proteins in the TP16 dataset, MurJ and PF0708 (PfmATE), were also examined in the initial MSA subsampling and SPEACH_AF papers. The authors should note that the diversity seen in this work is significantly less than in the previous presentations.
3. The legend to Fig. 6 is incorrect and should be corrected.
4. The text and Figure 7 use both MSE and MAE interchangeably, and the authors should ensure consistency in the terminology.

Reviewer #2

(Remarks to the Author)

The authors have successfully addressed my original concerns. Figure 5 is a nice addition to the manuscript as it shows a good agreement and similarity between the predicted structures and experimentally validated structures, thus providing further evidence of the validity of the approach. In my opinion the paper can be accepted without any additional change.

Reviewer #3

(Remarks to the Author)

I agree with the publication of the paper, but in addition to deep learning models, optimization methods are also worth considering in the protein multi-conformation prediction problem, such as [1] and [2].

[1] Peng C, Zhou X, Liu J, et al. Multiple conformational states assembly of multidomain proteins using evolutionary algorithm based on structural analogues and sequential homologues[J]. Fundamental Research, 2024.

[2] Hou M, Jin S, Cui X, et al. Protein multiple conformation prediction using multi-objective evolution algorithm[J]. Interdisciplinary Sciences: Computational Life Sciences, 2024: 1-13.

Response to reviewers

We would like to thank the reviewers for their insightful comments. Based on their comments, we have substantially revised the manuscript. All the changes are highlighted in red in the manuscript, and point-to-point answers to the reviewer's questions are provided below.

In addition, during the revision, we noticed that the MSAsubsample implemented in AFcluster that we had used was significantly different compared to the original reference (del Alamo, 2022) for MSAsubsample. We reimplemented MSAsubsample following the original reference since there is no code in their git repo to run MSAsubsample standalone. This new version of MSAsubsample performs significantly better than the suboptimal version in AFcluster. This explains why all the results related to MSAsubsample have changed in the revised manuscript. I am really happy that we spotted this and could correct it.

Cheers

Björn Wallner

Reviewer #1 (Remarks to the Author)

This manuscript describes another method, AFSample2, aimed at increasing model diversity with AlphaFold2. It compares results from the new method to three other methods that have been proposed to also increase diversity and the base AlphaFold2 method. In addition, they propose a new method to identify alternate protein conformations. This method is a derivation of the previously published method, SPEACH_AF. While the results presented are of interest, it is unclear why they propose a new method for comparing models without using ground truth structures when principal component analysis (PCA) is a widely used method to examine conformational diversity in the absence of ground truth structures. In addition, I feel the improvement over AFvanilla is overstated.

[Response] Thanks for your comments; we have updated the manuscript accordingly and provided point-by-point answers to the specific questions below.

1. The manuscript should clarify the exact number of cases where AFSample2 improves upon AFvanilla. Of the 17 cases in the OC23 set where AFSample2 improves the TMscore of the open conformation, 7 of these already yield both conformations with AFvanilla based on the TM-score>0.8 criteria and 3 of these do not generate models that cross the 0.8 threshold, yielding 7 cases of having both an improvement over AFvanilla and yielding both conformations. "On the other hand, AFsample2 is able to successfully generate high-quality models of both states without failing for even a single target in the OC23 dataset." There appear to be three cases, P00558, Q9ERE7, and P62495, where both conformations do not achieve a TM-score>0.8.

[Response] We have updated Figures 3 and 9 to make the improvements over AFvanilla clearer. In particular, we are now also showing the minimum TM-score to the two reference states (Fig 3c, 9c), this measure captures the importance of being close to both states and is easier to visualize than both open/closed similarities in scatter plot. In this measure, the improvement over AFvanilla is clear, AFsample2 improves ($\Delta TM > 0.05$) in 9 cases on OC23 and 11 in TP16, and it is never worse (Figures 3c and 9c).

Thanks for pointing out that “*On the other hand, AFsample2 is able to successfully generate high-quality models of both states without failing for even a single target in the OC23 dataset.*” This statement was obviously wrong, and we have updated the text to “*for most targets in the OC23 dataset*”.

2. Concerns regarding the RMSF analysis and presentation.

2a. The assumption is that the $\Delta C\alpha$ are the wings of the normal distribution for the disorder of the models, RMSF, spanning from open to close. The correspondence between the calculated slope/Z-score and the expected value needs further explanation, especially for cases where both conformations are not achieved, the model distribution is not normally distributed, or the fill factor is low.

[Response] Thank you for bringing this to our attention. We agree that the assumptions may not hold in all cases, particularly when both conformations are not achieved. As a result, we have decided to significantly revise this section, removing much of the previous analysis, including the results related to the slope and Z-score. Given that the relationship between $\Delta C\alpha$ and RMSF can vary across proteins, it is challenging to define a universal benchmark for what constitutes an appropriate level of RMSF.

We have replaced this section with a more concise discussion that focuses on $\Delta C\alpha$ and RMSF line and scatter plots, correlations, and RMSF distributions across different $\Delta C\alpha$ ranges. This revised analysis highlights that the ensembles generated by AFsample2 exhibit greater conformational diversity compared to those from AFvanilla. In particular, in regions where the $\Delta C\alpha$ is high.

2b. The legend should clearly indicate the meaning of the red lines in Figs 5b-f and S4. If they are the fit lines, why do some of them not appear to match the data very well?

[Response] Thanks for this comment. We have removed this analysis from the manuscript.

2c. The reason for the split data in some of the plots should be discussed.

[Response] Thanks for this comment. We have removed this analysis from the manuscript. But the split was most likely a problem with superposition.

3. “Identify protein states without ground truths”.

3a. A comparison of this new method to PCA should be carried out.

[Response] We have toned the novelty of this “new method”. It was really not meant to be a replacement for PCA or regular clustering. The idea of the new method was to provide a simple yet effective way of *selecting* two states. With PCA, you project an ensemble, usually on the first two components, which is useful if you want to visualize known states on that projection, but there is, in general, no physical meaning of these components that can guide you in selecting two states from the PC1-PC2 projection. In addition, sometimes, the separation of points can be misleading if the variation in the data is not coupled with the changing of states. The new method does not replace PCA; it simply monitors the model confidence as a function of the similarity to one of the predicted states (selected to be the highest confidence model), with the goal of identifying high-confidence models that are not similar to the predicted state.

We have also clarified this in the manuscript, and we also provide a comparison to PCA and examples of why it is not straightforward to predict states from a PCA projection in the revised manuscript (see section: *Selecting protein states* and Figure S5). We also provide PCA projections for all ensembles alongside the “new method” in Figure S6.

3b. The reference structures used to determine the best possible selection should be clarified. Are they a crystal structure?

[Response] Yes, they are crystal structures. The best possible selections are the **two models** from the generated ensemble that are most similar to the two reference states (crystal structures). We have clarified this in the revised manuscript.

3c. The large gap between the optimal selection and the rest of the models in Figs. 6i, k and S8 needs explanation.

[Response] This ambiguity stems from the fact that the target sequences are taken from UniProt, and can contain flexible termini, or regions that are never similar in any model. This point has been clarified in the revised manuscript.

3d. The method's ability to return the open and closed states should be evaluated and discussed.

[Response] The method’s ability to identify good open and closed models from the generated ensemble without the aid of ground-truth crystal structures is discussed in the section *Selecting protein states* in the revised manuscript.

4. In the comparison with the Transporters set, the threshold for indicating AFSample2's success was set at 0.6, which is lower than the 0.8 threshold used for the OC23 set. This lower threshold should be explained. Additionally, the rationale for showing AFcluster in Fig. 7g should be provided, since in the original Xie and Huang paper, MSAsubsampling was reported to be better than AFcluster at obtaining both conformations.

[Response] In general, the transporters are more difficult targets for generating alternate states, which was the main reason for using a lower threshold for success. However, in the revised manuscript, we have now used $TM > 0.8$ consistently.

There is no rationale for showing AFcluster; we have removed the comparison and provide diversity plots for all targets and methods in Supplementary Figure S7 instead.

5. The comparison of AFSample2 to SPEACH_AF on the same set of proteins should be carried out.

[Response] Thanks for pointing. We have added SPEACH_AF in the benchmark.

6. The requested additional method details should be included in the manuscript.

6a. What were the parameters used for MSAsubsampling.

6b. What were the parameters used for AFcluster.

[Response]

We have added a description of how each method was run in the Methods section.

6c. Was dropout used for AFSample2? If so, wouldn't dropout potentially reduce the effect of column masking?

[Response] Yes, dropout was used. But we do not believe that the effect of column masking would be reduced. Since we are removing input information and adding dropouts on top would only generate more diversity. We are also performing random, as opposed to biased (like SPEACH_AF), column masking, and applying random on random would only make it "more" random.

7. The legends for Figures 3 and 5 are incomplete and should be updated.

[Response] Legends are updated

Reviewer #2 (Remarks to the Author):

Kalakoti and Wallner describe in this study a new approach named Afsample2 for predicting conformational ensembles with AF2. Afsample2 is benchmarked against many different previously reported strategies, and the results show a significant improvement when predicting additional states. Of relevance is the fact that Afsample2 was shown to work also with a case example of a IDP, which is interesting providing the inability of the other strategies. The paper is interesting. However, it should be much better stated and probably visually represented in the paper (i) how different Afsample2 is with respect to the previous approaches. I understand that Afsample2 randomly masks MSA positions to Ala, but this strategy was also applied in previous studies (for instance SPEACH_AF, and recently in 10.1021/acs.jctc.4c00222). (ii) It should also be better stated that Afsample2 also increases the sampling rate and introduces noise by enabling dropout layers at inference as AFsample. Therefore it should be much better stated what is unique of Afsample2 and how it differs to other approaches, otherwise it is hard to

judge the novelty of the approach. In addition to that, the paper misses some contributions to the field, such as the recent 10.1021/acs.jctc.4c00222, or 10.1002/pro.4426 in which different templates coming from either X-ray or MD data were included. Some of the included bioarxiv references need to be updated as well.

[Response] Thanks for the suggestions; we have updated the manuscript, including SPEACH_AF in all comparisons, as well as clarified the differences in more detail. We have also improved the description of the AFsample2, in particular, that we mask to 'X' and not Ala and that models are generated with dropout activated should now be clearer. We have also updated the new Fig 1 to include a visual description of the method to make it clear that each model is generated with a different masked MSA.

We have also added the suggested references and updated the BioRxiv references.

I have also the following comments:

1. In Figure 5, the RMSF for the different strategies is represented for one of the studied cases. RMSF coming from MD simulations is often similar to the experimental B-factors, although I agree there might be some deviations coming from the different crystallization and simulation conditions. Still, it is a good practice to compare RMSF with B-factors, and so I believe it would be nice to include the experimental B-factors and compare whether the flexibility trends are well reproduced.

[Response]

We thank the reviewer for their suggestion. While we acknowledge that small fluctuations observed in MD simulations may align with experimental B-factors, we do not expect the relatively large changes observed between the different states—and within the ensemble of models—to correlate well with experimental B-factors. Additionally, differences in crystallization conditions significantly influence these comparisons; for example, the median correlation between experimental B-factors for the 23 proteins in open and closed reference states (OC23) is only 0.53.

To further explore this, we calculated the correlation between experimental B-factors and the delta distance of residues (i.e., the change in residue positions between open and closed states). The resulting median correlation was only 0.3. Indicating that the B-factors do not agree with changes between open and closed states.

While we strive to incorporate more comparisons to experimental data, in this specific case, a comparison to experimental B-factors would not be an effective approach to demonstrate that the flexibility trends are well reproduced. Since not even the B-factor themselves agree with changes between open and closed states. We would be happy to discuss this further if needed.

2. As shown in Figure 4 and later figures, AFsample2 can provide open, closed and intermediate states of a given protein. The TM-score of the best open and closed models are computed. However, I wonder about the physical validity of the intermediate states. I understand that only structures with high pLDDT scores are taken, but still, in my opinion it would be nice to generate a PC space with all available PDB structures of the studied protein, and plot on this experimental X-ray based PC space the multiple open, closed and intermediate structures generated with Afsample2.

[Response] Thank you for the excellent comment. As suggested, we queried the PDB with sequences in the OC23 dataset (sequence similarity>90%) with the expectation of finding intermediate experimental structures. While no intermediates were found for most of the set, we found intermediate structures for P31133, Q7DAU8, P71447, and P33284. When mapped on the AFsample2 ensemble diversity plots, excellent agreement was observed between the similarity of experimental intermediates and ensemble intermediates (see Figure 5).

Reviewer #3 (Remarks to the Author):

The author introduces AFsample2 for predicting multiple conformations and ensembles with AlphaFold2. The author introduces AFsample2 for predicting multiple conformations and ensembles with AlphaFold2. This issue is very important for studying protein mechanisms. Here are some questions I hope the author can address:

1. The author samples different protein conformations by randomly masking the MSA and provides what they believe to be appropriate parameters: 1000 samples at 15% masking. I would like to know whether the author has conducted repeated experiments? Is there randomness involved? Especially given that the test dataset is not very large, is this conclusion reliable?

[Response] We have repeated the experiments and got the same results. When we sample different randomization in the ablations the trends are also as expected, with no abrupt deviations.

2. The complete protein dataset and the author's experimental results were not accessible through the provided link. Could the author provide the relevant data, including the predicted structures, to facilitate reproduction of the results?

[Response] Thanks for this comment, we have uploaded all relevant data to Zenodo, and a link to the data is provided in the manuscript and also in the git repo. We will include the DOI to the data when the work is published.

3. The datasets presented by the author, OC23 and Transporters, are relatively small, and the author might consider, further expanding them. In addition these test proteins seem to have only the final state structures. Is there an example that can provide a comparison between the real ensemble and the ensemble predicted by AFsample2?

[Response] We agree that having more data would be ideal, and we are working on a more extensive data set for future studies. However, it was possible to get significance for the improvements despite the small data size.

We could not find a real ensemble for the cases we tested, but we did find structures in the PDB that could be mapped to intermediate structures between the end states modeled by AFsample2 (see Figure 5).

4. The author provided a parameter of 15% masking in the paper. I would like to know the impact of random MSA masking on the performance of AF2 structure prediction.

[Response] AF2 structure prediction is surprisingly robust to MSA masking. In Fig 2a we show the ability of AF2 to predict the states as a function of the amount of MSA masking. Masking up to 40 % predicts the closed state, while MSA masking of 50% has a severe impact on the structure prediction. However, not for every case, for one target masking 50% was the best to generate one of the states.

5. AFsample2 predicts ensembles by random masking the input MSA. Previous similar works, such as changing the number of input MSAs, have shown that reducing the MSAs leads to a significant decline in the quality of the predicted structure. Could the author discuss whether modifying or reducing MSA information to predict ensembles is reasonable? After all, AF2 was originally designed to predict static structures, and ensembles were not part of the initial design framework.

[Response] Yes, it is reasonable. The rationale for our masking strategy is that the evolutionary information in the MSA contains overlapping evolutionary restraints which often results in a model with as many restraints as possible fulfilled. This is often the closed state. By masking columns, we break evolutionary restraints and allow the AF2 to explore structures fulfilling less restraints, i.e. alternate states. Of course, the ensembles will probably not be Boltzmann distributed, but are more a reflection of possible structures given the MSA

6. Has the author considered altering template information to predict ensembles?

[Response] Yes, for simplicity we choose not to use any templates in this study.

7. Do the predicted structures obtained by changing the input truly reflect the native structure? For example, are there cases where experiments do not show that the protein has multiple conformations, but the predicted structure is very different after changing the MSA?

[Response] No, structures do not change. We have added a test where we apply AFsample2 to proteins with varying differences between the states from TMscore 0.51 to 1.0. For proteins with large differences, the distribution of models is also large, and for proteins with no change, the distribution is very narrow (Fig S1). We have added a section about this in the manuscript since it is often overlooked.

8. How do authors determine whether a protein has multiple conformations? Or how to assess how many final state conformations a protein has?

[Response] The focus of this study was two-state conformations; we are certainly interested in looking at proteins with multiple conformations including intermediate states. It seems from our studies that it is possible to discriminate between single-state and multi-state at least. If a protein has three states and all are predicted with high-confidence it should be possible to use clustering or iteratively selecting high-confidence models that are not similar to previous selected models. But in general, it is difficult since the predictions are not perfect.

9. Has the author studied ensembles of complexes? Are there relevant datasets and experimental results that can be shared? For example, has AFsample2 been further extended to AF2-multimer?

[Response] Based on our limited internal testing with previous CASP complexes, AFsample2 works for multimers. However, the results are not extensive enough to make a claim. AF2-multimer is more complicated due to the importance of MSA pairing in getting to an accurate structure, and we might want to bias the sampling to the surface or potential chain-chain interactions. Given this, we aim to have a version of this method for multimers soon.

Response to reviewers

We would like to thank the reviewers for carefully assessing our revised manuscript and include the response to the final comments below.

Cheers

Björn Wallner

REVIEWERS' COMMENTS:

Reviewer #1 (Remarks to the Author):

The authors have made significant improvements based on the initial reviews. The comparison to the other methods supports the utility of this methodology. Even if not entirely novel, the ability to yield more diverse conformations and additional intermediates for even one example is worth publishing. However, there are still a few points that should still be addressed:

- 1. The statement about SPEACH_AF relying on having experimental structures or predictable models for one of the states is true, but the authors should note that AFvanilla is able to generate one of the conformations in all cases, so this argument seems spurious as SPEACH_AF uses an unmodified MSA. It is also surprising that no score is reported for A6UVT1, as the unmodified MSA should yield a viable**

conformation.

[Response] It is not a given that AFvanilla will predict one of the states in every single case, even though it does for OC23, but we already mentioned that “SPEACH_AF relies on having [...] models that can be predicted for one of the states”. Indeed, that is also how we run SPEACH_AF.

We have investigated the target A6UVT1, it seems the reason for the no score is that the SPEACH_AF code crashes using the starting model generated by AFvanilla. We tracked the problem to improper exception handling of empty lists, corrected it, and now SPEACH_AF has results for A6UVT1.

- 2. Two of the proteins in the TP16 dataset, MurJ and PF0708 (PfMATE), were also examined in the initial MSAsubsampling and SPEACH_AF papers. The authors should note that the diversity seen in this work is significantly less than in the previous presentations.**

[Response] Thank you for bringing this to our attention. We reported fantastic performance for MurJ and PF0708 (**Fig 8** and **Fig S2**) after the first round of revision for MSAsubsampling. However, we failed to update the diversity plots in the revised supplementary (**Fig S7**), leading to the mismatch. We have now updated the supplementary plots.

For SPEACH_AF, we re-ran MurJ and PF0708 and observed very similar results. Unfortunately, SPEACH_AF does not have any code nor data (MSA and seed model) to reproduce the reported results.

- 3. The legend in Fig. 6 is incorrect and should be corrected.**

[Response] Corrected

- 4. The text and Figure 7 use both MSE and MAE interchangeably, and the authors should ensure consistency in the terminology.**

[Response] Thanks for pointing this out. It should be MAE, and the relevant text has been corrected.

Reviewer #2 (Remarks to the Author):

The authors have successfully addressed my original concerns. Figure 5 is a nice addition to the manuscript as it shows a good agreement and similarity between the predicted

structures and experimentally validated structures, thus providing further evidence of the validity of the approach the paper can be accepted without any additional change.

[Response] Thanks, we also like Figure 5!

Reviewer #3 (Remarks to the Author):

I agree with the publication of the paper, but in addition to deep learning models, optimization methods are also worth considering in the protein multi-conformation problem, such as [1] and [2].

[1] Peng C, Zhou X, Liu J, et al. Multiple conformational states assembly of multidomain proteins using evolutionary algorithm based on structural analogues and sequential homologues[J]. Fundamental Research, 2024.

[2] Hou M, Jin S, Cui X, et al. Protein multiple conformation prediction using multi-objective evolution algorithm[J]. Interdisciplinary Sciences: Computational Life Sciences, 2024: 1-13.

[Response] Indeed, unfortunately, they don't compare to any of the AlphaFold sampling methods.